# Exploring Asynchronism in SWARM Parallelism

**Yan Zuo, Gil Avraham, Thalaiyasingam Ajanthan, Sameera Ramasinghe & Alexander Long**
Pluralis Research
{yan,gil,aj,sameera,alexander}@pluralis.ai

## Abstract

SWARM parallelism is a framework that enhances pipeline parallelism in distributed training by incorporating fault tolerance. However, the synchronous nature of this approach introduces inefficiencies that can hinder performance and scalability. We analyze these inefficiencies and propose an asynchronous modification to the framework that enables nodes to perform local updates and periodically average their states. Our results demonstrate that this modified asynchronous SWARM achieves higher throughput without sacrificing model convergence.

## 1 Introduction

Collaborative training enables large neural networks to be trained using distributed computational resources. Frameworks like DiLoCo Douillard et al. (2023) and DiPaCo Douillard et al. (2024) support this across geographically dispersed nodes but are limited to data parallelism. SWARM Parallelism Ryabinin et al. (2023) extends this by incorporating pipeline parallelism Huang et al. (2019), allowing contributors with consumer-grade accelerators to participate while maintaining node elasticity.

To preserve centralized training dynamics, SWARM accumulates gradients within pipeline stages and performs periodic all-reduce operations. However, this introduces synchronization bottlenecks that worsen as the number of stages increases. Asynchronous execution could improve efficiency but introduces gradient staleness, disrupting training dynamics and degrading convergence. This paper analyzes gradient staleness in the asynchronous setting of SWARM and proposes a weight correction technique using Nesterov Accelerated Gradient (NAG) Nesterov (1983; 2013). Experimental results demonstrate its effectiveness in improving training stability and efficiency.

## 2 Method

Introducing asynchronous execution in SWARM can be approached along two axes: (1) enabling nodes within a stage to perform local updates, and (2) performing state averaging within a stage without halting the training loop. In this work, we focus on the first approach and leave the latter for future research. Allowing local updates inevitably introduces *gradient staleness*, a well-documented issue in asynchronous pipeline-parallel training Narayanan et al. (2019); Yang et al. (2021). In the SWARM framework, which supports heterogeneous environments, an additional dimension of staleness arises as nodes with lower backward-pass throughput accumulate compounding delays. The gradient staleness in an asynchronous pipeline-parallel setup can be formally expressed as:

$$\mathbf{w}_i^{t+1} = \mathbf{w}_i^t - \eta \, \nabla f_i(\mathbf{w}_i^{t-\tau_i}) \tag{1}$$

where $\tau_i$ represents the delay in gradient updates, which increases with the stage depth in the pipeline. In the case of SWARM parallelism in a heterogeneous setting, stochastic rewiring—a key mechanism for optimizing bandwidth utilization—introduces an additional stochastic staleness factor $\tau_i$, as mini-batches traverse the network in a non-deterministic manner. Large language models (LLMs) Brown et al. (2020) are typically trained using AdamW Loshchilov (2017), but in this work, we explore an alternative approach based on Nesterov Accelerated Gradient (NAG):

$$\mathbf{d}_t = \gamma_t(\mathbf{w}_t - \mathbf{w}_{t-1}) \, , \tag{2}$$
$$\mathbf{w}_{t+1} = \mathbf{w}_t + \mathbf{d}_t - \eta \nabla f(\mathbf{w}_t + \mathbf{d}_t)$$

Unlike standard momentum, which applies a velocity term to past gradients, NAG adjusts the update direction before computing the gradient, reducing overshooting and improving stability Nesterov (1983; 2013). In our context, the velocity component $\mathbf{d}_t$ acts as a weight correction term, compensating for gradient staleness by anticipating the future weight position. This allows NAG to mitigate stale updates by dynamically adjusting the weight trajectory based on past updates and estimated future gradients.

## 3 RESULTS

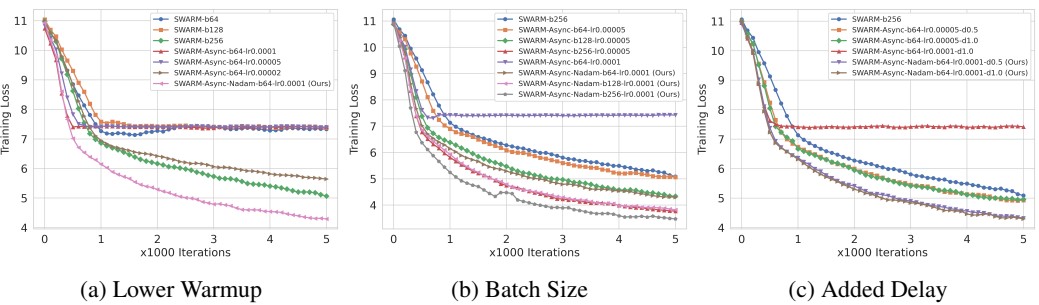

|            (a) Lower Warmup            |            (b) Batch Size            |            (c) Added Delay            |

Figure 1: Training trajectory comparisons of SWARM and its asynchronous variants. Our method significantly outperforms competing baselines across various ablations and significantly stabilizes training dynamics.

For our SWARM experiments, we use an 8-stage pipeline with two worker nodes per stage, training on the WikiText dataset. The asynchronous variant of SWARM relaxes strict synchronization constraints, allowing local updates per microbatch with periodic stage-wise weight synchronization. Further details on our setup are provided in Appendix A. We evaluate three configurations: (1) standard synchronous SWARM, (2) asynchronous SWARM with periodic weight synchronization (SWARM-Async), and (3) our NAG-adapted SWARM-Async.

As shown in Fig. 1, our method consistently outperforms both baseline variants across different conditions. Fig. 1a highlights that SWARM-Async struggles to converge when warmup time is halved to 500 iterations, whereas our NAG-adapted approach remains stable under this setting. Similarly, Fig. 1b illustrates that SWARM-Async converges at a slower rate when compared to our method, which tolerates more aggressive local updates. In Fig. 1c, we introduce artificial delays (0.5s and 1.0s) in one node per stage, forcing stochastic routing to favor non-delayed nodes and increasing gradient staleness. Our NAG-adapted approach proves more resilient to these disruptions, significantly outperforming the baselines in terms of stability and convergence. Additionally, Tab. 1 compares relative wall-clock times between synchronous and asynchronous SWARM, where we observe a noticeable improvement in relative wallclock time in asynchronous SWARM.

## 4 CONCLUSION

We explore the asynchronous setting of SWARM parallelism and introduce a novel variant of Nesterov Accelerated Gradient (NAG) to counteract gradient staleness and stabilize training. Our approach requires only a simple optimizer switch and a minor hyperparameter adjustment, making it easily adoptable with immediate benefits in robustness and convergence. Empirically, it consistently outperforms competing baselines, maintaining strong performance in decentralized SWARM settings. This work highlights the potential of

|                | **Wallclock** |
|----------------|---------------|
|                | Rel. Improvment |
| Batch Size 64  | 45.9%         |
| Batch Size 128 | 32.6%         |
| Batch Size 256 | 23.4%         |

Table 1: The relative improvement in wall-clock time for each method across 5000 iterations for different weight synchronization batch sizes between synchronous and asynchronous SWARM

lightweight optimization adjustments to improve asynchronous training, paving the way for more efficient and scalable collaborative learning frameworks.

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

## A    SWARM SETTINGS

We adopt the SWARM baseline from Ryabinin et al. (2023) for our decentralized training framework. The model used across all baselines is a Transformer language model with an architecture similar to prior work Brown et al. (2020); Wang & Komatsuzaki (2021).

Our SWARM setup consists of 2 worker nodes per stage, totaling 16 worker nodes, each equipped with an NVIDIA L4 GPU. Additionally, we allocate 16 trainer nodes to manage the full pipeline, where each trainer node features a 4-core Intel Cascade Lake CPU (2.2 GHz base clock) and 32 GB of RAM.

For all baselines, we use the following model configuration:

- Embedding dimension: 768
- Number of attention heads: 6
- Feedforward layer dimension: 3072
- Number of layers: 8 (each assigned to a separate pipeline stage)
- Microbatch size: 8
- Sequence length: 2048

We set the base learning rate to $1e\text{-}4$ for all baselines including our Nesterov-based approach. By default, all models use a linear warmup schedule up to 1k steps. Our Nesterov-based approach uses a momentum coefficient $\beta_1$ set to 0.99 with a learning rate of $1e\text{-}4$. All models are trained for 5k iterations, using a stage-wise all-reduce batch sizes of 64, 128 and 256.

## B    ADDITIONAL RESULTS

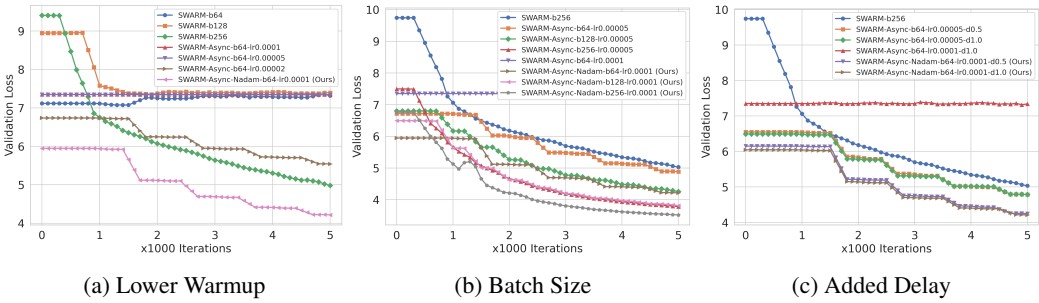

| (a) Lower Warmup | (b) Batch Size | (c) Added Delay |

Figure 2: *Validation trajectory comparisons of SWARM and its asynchronous variants. Our method significantly outperforms competing baselines across various ablations*

