# OpenReview forum: "Exploring Asynchronism in SWARM Parallelism"
_ICLR.cc/2025/Workshop/MCDC — MCDC @ ICLR 2025_

### Official Review · Reviewer_h9q3 · 2025-02-26

**Rating:** 7
**Confidence:** 4
**Fit:** 5

**Summary:**

In this work, Nesterov Accelerated Gradient is combined with SWARM parallelism to improve performance of asynchronous SWARM parallelism. Inspiration is drawn from papers such as Async Local-SGD (Liu et. al. 2024) to allow for increased hardware utilization.

**Reason For Giving A Higher Score:**

Strong ideas and fundamentals. Well aligned with workshop goal of modularity. Convincing experimental results

**Reason For Giving A Lower Score:**

Sparse explanation, leaving a lot up to the imagination.

**Strengths And Weaknesses:**

The concept behind this paper is compelling; to combine a DiLoCo-style distributed training technique with pipeline parallelism. Pipeline parallelism (via SWARM) allows larger models to be trained, whilst DiLoCo reduces communication requirements between nodes. The paper notes the challenge of gradient staleness when applying an inner-outer training technique such as DiLoCo, and is convincing in its application of Nesterov Accelerated Gradient for alleviating these issues.

The experiments provided were compelling, showing the strength of the combined async+NAG training strategy.

However, I felt there was a lack of explanation of some of the key ideas in this paper. The method was not clearly explained and left quite a lot to the imagination. Further elaboration of the method, algorithm, and rationale would be beneficial.

**Suggestions:**

As mentioned above, whilst this paper targets a very interesting idea, I felt that some of the explanations were quite sparse. I would advise elaboration on the following:

- Explain the previous work on this field. DiLoCo is crucial here, and there are works on combining this with SWARM (eg DiLoCo-SWARM, Mika Senghaas, 2025)
- Explain how DiLoCo or DiPaCo are implemented in your work. I can't tell from your paper whether you used either of them - if you didn't use them it should be clear how they are relevant and why they weren't used. In general a more full explanation of your method would be beneficial.
- Explain where the 'async' step comes in. Maybe an written-out algorithm would be informative.
- Explain why gradient staleness occurs

---

### Official Review · Reviewer_cmvd · 2025-02-27

**Rating:** 7
**Confidence:** 3
**Fit:** 4

**Summary:**

The paper extends SWARM Parallelism, a distributed training framework that allows geographically dispersed nodes with consumer-grade accelerators to participate in collaborative training. It addresses the synchronization bottlenecks caused by gradient accumulation across pipeline stages by introducing an asynchronous execution strategy. However, asynchronous updates can lead to gradient staleness, which degrades convergence. To mitigate this, the authors propose a weight correction technique using Nesterov Accelerated Gradient (NAG). Their experiments demonstrate that this approach improves training stability and efficiency in asynchronous settings, making distributed training more scalable and resilient.

**Reason For Giving A Higher Score:**

N/A

**Reason For Giving A Lower Score:**

N/A

**Strengths And Weaknesses:**

Strengths

* The paper provides experimental validation through multiple ablation studies, demonstrating the effectiveness of the proposed approach under different conditions (e.g., varying batch sizes, warm-up times, and artificial delays).

* The results indicate that the NAG-adapted SWARM approach significantly improves training stability and convergence compared to both synchronous and asynchronous baselines.

* The asynchronous execution strategy relaxes synchronization constraints, allowing for more efficient training with reduced wall-clock time.

Weaknesses
* The paper focuses solely on enabling local updates within pipeline stages but does not explore state averaging techniques, which could further enhance performance.

**Suggestions:**

* Demonstrating how performance scales with number of pipeline stages or nodes would make the results more impactful

* "In our context, the velocity component dt acts as a weight correction term, compensating for gradient staleness by anticipating the future weight position" - Would be interesting to perform an analysis to determine whether this is empirically observed.

---

### Official Review · Reviewer_m9cB · 2025-02-28

**Rating:** 7
**Confidence:** 5
**Fit:** 5

**Summary:**

The work studies SWARM parallelism, a distributed parallelism technique that is 1-step-stale synchronous, but the paper studies how SWARM parallelism works with local asynchronous updates which have the potential to reduce bandwidth requirements of training considerably. The paper finds that Nesterov's accelerated gradient stabilizes asynchronous training.

**Reason For Giving A Higher Score:**

This is a great workshop paper that will lead to discussion among its participants.

**Reason For Giving A Lower Score:**

The finding is neat, but compared to other work might not have a wide impact because distributed training over multiple regions is still a relatively uncommon way of training large models. As such, this is solid work that is more niche and with that only interesting to some sub-population in the workshop. Other work might have larger impact.

**Strengths And Weaknesses:**

The paper studies a simple hypothesis and presents a simple and effective solution. This is the hallmark of a good workshop paper.

The weaknesses are mostly that more extensive hyperparameter search, particularly for the baselines, and more extensive experiments would strengthen the paper -- but this is also a usual mark of a workshop paper.

**Suggestions:**

I think more diverse experiments and careful comparison under more hyperparameter settings are usual. Beyond that, mostly asynchronous approach break down with scale, so training on a larger scale would be important for a full paper.

---

### Decision · Program_Chairs · 2025-03-06

**Decision:**

Accept

**Comment:**

This work proposes an asynchronous version of the SWARM pipelining. Enabling inference of model across distributed devices is a relevant topic to this workshop, and all reviewers recommend acceptance, therefore we're please to accept this paper to the workshop.